# How General and Inflammatory Status Impacts on the Prognosis of Patients Affected by Lung Cancer: State of the Art

**DOI:** 10.3390/biomedicines12071554

**Published:** 2024-07-12

**Authors:** Antonio Mazzella, Riccardo Orlandi, Sebastiano Maiorca, Clarissa Uslenghi, Matteo Chiari, Luca Bertolaccini, Monica Casiraghi, Giorgio Lo Iacono, Lara Girelli, Lorenzo Spaggiari

**Affiliations:** 1Division of Thoracic Surgery, IEO, European Institute of Oncology IRCCS, 20141 Milan, Italy; riccardo.orlandi@unimi.it (R.O.); sebastiano.maiorca@unimi.it (S.M.); clarissa.uslenghi@unimi.it (C.U.); matteo.chiari@ieo.it (M.C.); luca.bertolaccini@ieo.it (L.B.); monica.casiraghi@ieo.it (M.C.); giorgio.loiacono@ieo.it (G.L.I.); lara.girelli@ieo.it (L.G.); lorenzo.spaggiari@ieo.it (L.S.); 2Division of Oncology and Hemato-Oncology, University of Milan, 20141 Milan, Italy

**Keywords:** lung cancer, inflammation, inflammatory status, general status, metabolism, metabolic status, prognosis, NSCLC

## Abstract

Pulmonary cancer is often associated with systemic inflammation and poor nutritional status and these two aspects are strongly correlated and related to the scarce infiltration of a tumor by immune cells. We reviewed all English literature reviews from 2000 to 2024 from PubMed, Scopus and Google Scholar, including original articles, review articles, and metanalyses. We excluded non-English language articles and case reports/case series. Generally speaking, nutritional and inflammatory status largely affect medium and long-term prognosis in lung cancer patients. A correct stratification of patients could improve their preoperative general functional nutritional and inflammatory status, minimizing, therefore, possible treatment complications and improving long-term prognosis.

## 1. Introduction

Lung cancer is actually the second most commonly diagnosed cancer in both men and women; nevertheless, it represents the leading cause of cancer-related deaths in males and females, causing, in 2020, more deaths than breast, colorectal, and prostate cancers combined. In 2023, the notable number of new cases of lung and bronchus cancers in the US was 238,340; the estimated deaths in the same year for lung and bronchus cancers was 127,070. In other words, this significant figure almost represents a health emergency, being the leading cause of death in both men and women [1,2].

Much has been investigated about the risk factors, staging, prognosis, and treatment of lung cancer; in addition, in the last few years, we have been witnessing the notable progress of molecular medicine for the tailor-made treatment of lung cancer (i.e., biologic therapy). The assumption that it is not necessary to cure the disease as such, but rather treat the patient suffering from the disease, has become increasingly fitting. In light of these circumstances, studying every general, molecular, metabolic, physical, and inflammatory aspect in the status of these patients becomes of paramount importance.

Pulmonary cancer is often associated with systemic inflammation and poor nutritional status; indeed, these two complementary and consequential aspects are features of lung cancer, both in patients with limited and oligometastatic disease [3,4,5,6,7], and they are strongly correlated and related to scarce infiltration of tumors by immune cells [5]. Inflammatory status (pre-existent or concomitant with lung cancer), with subsequent increased energy consumption, contributes to malnutrition and vice versa. In cancer patients, we normally observe catabolic processes (secondary to inflammation), reduced caloric intake, fat and muscle loss [4,5], and an imbalance in anabolic and proteolytic pathways [8,9,10].

Thus, inflammation status, general status, and nutritional status are inversely correlated and their imbalance may be responsible for a significant loss of muscle tissue, associated with a worse prognosis.

Accordingly, the general health status of a cancer patient and his/her response to surgical/medical treatment could be determined, other than by the illness, by these host-specific interrelations. In other words, the patient’s “fitness” (nutritional status, systemic inflammation, nutritional status, muscle mass/fat distribution, endurance tolerance, and ventilatory reserve) could be a prognostic determinant in the short and long term.

In this paper, we will try to understand how the host-specific status can influence the prognosis of patients suffering from lung cancer, by evaluating all the data currently present in the literature.

We will focus on the inflammatory, metabolic, and nutritional status of the patients, trying to understand how they can positively or negatively influence cancer prognosis.

## 2. Materials and Methods

### 2.1. Research Selection

An extensive literature search was conducted on 20 March 2024, across reputable databases (PubMed/Medline, Scopus, and Google Scholar). We used a variety of search terms/keywords to identify relevant studies. These terms included the following: “lung cancer”, “systemic inflammation”, “inflammation”, “inflammatory status”, “general status”, and “nutritional status”. A comprehensive description of the search strategies is illustrated in Table 1. In addition, we manually screened the reference lists of the found studies in order to include other significant articles. We finally performed research using Google Scholar, cross-referencing the results and adding further works not indexed on PubMed/Medline.

### 2.2. Inclusion/Exclusion Criteria

We included in this review all original articles, review articles, and metanalyses published between 2000 and 2024.

We excluded non-English language papers, case reports, and case series.

## 3. State of the Art

The relationship between cancer, general status, and inflammation has been largely discussed in the literature [11,12,13,14,15]. From the data extracted from our research, we found several different parameters able to influence the development and prognosis of patients affected by lung cancer.

### 3.1. General and Nutritional Status: Physical Fitness

In the general population, “physical fitness” is considered the ability to carry out daily tasks with vigor and alertness [16]. In patients affected by lung cancer, the concept of “fitness” may be considered differently. The general and nutritional status of oncologic patients is generally evaluated by different clinical, biological, and anthropometric parameters, particularly BMI (body mass index), body surface area, weight and weight loss, and “skeletal muscular status” (evaluation of skeletal muscular tissue) [3,4,5,6,8,9,10,17,18,19,20].

Pre-operative assessment, nutritional status, and general status are key features in the development of tumor characteristics and prognosis in patients affected by non-small cell lung cancer (NSCLC). The body distribution of fat and muscle is essential; many reports showed that BMI impacts survival independently from TNM [20] (Table 2).

A 2014 report by *The Lancet* shows how obesity may increase the risk of 13 different cancer types, not including lung cancer [21]. Nowadays, the impact of obesity on the long-term survival of cancer patients is really debated; in some cases, we use an oxymoron, called “the obesity paradox” [18,19,20,22,23,24], which demonstrates that patients with a pre-surgery BMI of <25 kg/m^2^ or >25 kg/m^2^ have a significant difference in 5-year survival. The “obesity paradox” of NSCLC may therefore apply to lung cancer risk occurrence [25], possibly due to the lower concentration of polycyclic aromatic hydrocarbons in lung cells due to carcinogenic molecule retention in fat tissue [26]; on the other hand, a higher BMI would also have a protective effect on the post-treatment outcome of lung cancer.

Another important key feature is weight loss, often associated with subsequent sarcopenia; this aspect may show a catabolic metabolism before the treatment and it is notably related to morbidity/mortality in hospitalized patients, especially if surgically treated [27,28,29]. It is also a predictive factor of shorter survival in patients with advanced non-operable lung cancers [30,31]. As Alifano [20] showed, weight loss impacts the long-term survival of all patients, regardless of TNM, and particularly those with a lower BMI.

Systemic inflammation and impairment of nutritional status are frequent features of lung cancer [3,4]. These two factors are strongly correlated in patients undergoing lung cancer surgery and both are related to the scarce infiltration of tumors by immune cells [5]. Inflammatory status (pre-existent or concomitant with lung cancer), with subsequent increased energy consumption, contributes to malnutrition. In turn, catabolic processes secondary to inflammation, and reduced caloric intake, may be responsible for fat and muscle loss [4,5]. Other mechanisms have also been invoked, including an imbalance in anabolic and proteolytic pathways [8,9].

Sarcopenia represents another cornerstone for the general and nutritional status of patients. Low muscle mass negatively impacts the survival of operated lung cancer patients [19,32]. Low muscle mass was mostly observed in underweight and normal weight patients (66% and 39%, respectively), less frequently in overweight patients (21%), and rarely in obese patients (9.5%) [20]. This scarcity of sarcopenic obesity (SO) in stage I–III patients agrees with the report of Prado et al. [33], which recorded 8% (20/250) of SO in a subpopulation of non-metastatic cancer patients, much lower than figures (around 25%) usually recorded in advanced and metastatic disease, where low muscle mass is included in a process of cachexia linked to poor survival [28,34]. It is noteworthy that muscle loss occurring in follow-up negatively impacts the survival of early-stage resected NSCLC cancer patients independently [35]. However, it is impossible to say if sarcopenia is likely to be a direct determinant of prognosis, but it is an extremely powerful marker of patient frailty.

More generally, in cancer patients sarcopenia is associated (if not in part responsible for) physical decline, loss of quality of life, increased toxicity from chemotherapy, and shorter survival. The combination of sarcopenia and cachexia is believed to be the direct cause of death in at least 20% of oncologic patients with advanced disease [9].

### 3.2. Inflammatory Status

There is a close interconnection between cancer development and clinical, general, and inflammatory status. Cho et al. [13] reported that cancer arises more easily in chronically inflamed tissues. There is a close connection between inflammation and lung cancer. The inflammatory molecules may be responsible for augmented macrophage recruitment, delayed neutrophil clearance, and an increase in reactive oxygen species. Chronic pulmonary disorders and the increased release of cytokines and growth factors represent the cornerstone for epithelial-to-mesenchymal transition and the constitution of a tumor microenvironment. A plethora of other studies have demonstrated the association of inflammation and poor prognosis of lung cancer [36,37,38,39,40,41,42,43,44,45,46,47,48].

Currently, inflammatory status evaluation is performed by dosing various parameters in the blood, before or during treatment (Table 3).

#### 3.2.1. C-Reactive Protein

C-reactive protein (CRP) is notoriously a marker of acute-phase inflammatory response. Nowadays, the reasons why high CRP levels are associated with worse prognosis in cancer patients are still unclear and remain topics of debate. Baseline CRP levels have been studied as a prognostic factor in early-stage NSCLC in multiple studies. One of the most recent was a meta-analysis showing that a baseline high CRP level is significantly associated with poor prognosis in early-stage NSCLC or in patients undergoing pneumonectomy for lung cancer [49,50]. CRP is not specific to cancer and its impact on survival is not limited to neoplastic diseases, but is observed in cardiovascular disease, osteoporosis, diabetes mellitus, COPD (chronic obstructive pulmonary disease), and generally in all age-related diseases [51]. Associations between high PCR levels and prognosis in lung cancer have been investigated, both in surgery [49,50] and other treatments [52]. All studies report the negative impact of a high preoperative PCR level on survival.

#### 3.2.2. Blood Count

Concerning clinical and inflammation status, preoperative complete blood count, albumin, and pre-albumin have been widely studied as promising prognostic predictors in some tumors, like hepatocellular carcinoma, gastric cancer, and lung cancer [15,36,37,38,39,40,41,42,43,44,45,46,47].

The most common abnormalities, linked to the chronic inflammatory process accompanying neoplasia, are represented by leukocytosis, neutrophilia, thrombocytopenia, and lymphocytopenia. They may occur during the growth and lysis of the tumor [15,45,53]. Immunoregulatory cytokines secreted by inflammation facilitate the recruitment of tumor-associated neutrophils, causing disease progression and increasing the risk of distant metastasis; platelets also appear to have a similar role. Conversely, lymphocytes are believed to have anticancer activity; indeed, lymphocytosis itself is considered a favorable prognostic factor [54].

Platelet count was found to be increased in lung cancer and colorectal cancer, which indicated poor survival outcomes [37]. Lymphocytes play an important role in the defense against cancer by inducing cytotoxic cell death and by inhibiting the proliferation and migration of cancer cells [38]. Hemoglobin and albumin are two of the most common indexes that reflect the performance and nutritional status of patients. Hemoglobin has been reported as a prognostic factor in cancer patients, and anemia has been associated with poor prognosis [39]. Albumin was also demonstrated as a prognostic factor in GC, revealing that patients with a higher level of albumin had better prognoses than those with lower levels [40].

Actually, referring to blood count and blood tests, we can find several different indexes for assessing the inflammatory status of patients at the moment of diagnosis/before surgical treatment and after surgery. The most relevant in the literature are as follows:The platelet-to-lymphocyte ratio (PLR) and albumin multiplied by lymphocytes, known as the prognostic nutritional index (PNI);The HALP amalgamated index, which is measured as hemoglobin (g/L) × albumin (g/L) × lymphocyte (/L)/platelet (/L);The serum polymorpho-nuclear neutrophil-to-lymphocyte ratio (dNLR);The lymphocyte-to-monocyte ratio (LMR): serum lymphocytes/monocytes;The systemic immune-inflammation index (SII): serum platelets * neutrophil/lymphocytes;The advanced lung cancer inflammation index (ALI): serum albumin * BMI/NLR; BMI = weight (kg)/height (m)^2^.

#### 3.2.3. HALP

In recent studies, a new composite index named HALP, calculated as hemoglobin (g/L) × albumin (g/L) × lymphocyte (/L)/platelet (/L), was reported to be related to survival in gastric cancer, colorectal cancer, bladder cancer, and renal cancer patients [15,46,47,48,55]. Chen et al. demonstrated that the HALP index was associated with tumor size and T stage. Low HALP was significantly associated with tumor progression and acted as an adverse prognostic factor in gastric cancer [11]. A recent paper [15] corroborates this aspect, testifying to the strong association between low HALP levels and bad prognosis in a homogeneous cohort of patients undergoing lobectomy.

However, recognized inflammatory markers associated with cancer prognosis have not been identified, and the association between inflammatory markers and clinical characteristics is poorly understood.

#### 3.2.4. Derived Blood Count/Serum Indexes (dNLR, PLR, LMR, NLR)

Recent studies have assessed the derived neutrophil–lymphocyte ratio (dNLR) as a novel serum inflammatory marker in patients with advanced or recurrent NSCLC, treated with molecular targeted therapy or immunotherapy [56,57,58,59,60,61]. Other studies have proposed a panel of inflammatory markers; albumin (Alb), C-reactive protein (CRP), lactate dehydrogenase (LDH), neutrophil–lymphocyte ratio (NLR), lymphocyte–monocyte ratio (LMR), platelet–lymphocyte ratio (PLR), and dNLR [15,62]. The authors identified LMR as an independent prognostic factor for DFS and OS in non-small cell lung cancer patients. The important role of cancer-specific cytotoxic T-cells for anticancer response has been previously reported; here, in addition, the authors discussed the function of tumor-associated macrophages for promoting tumor angiogenesis. However, their subset analyses revealed that LMR had a stronger impact on survival among patients with the following characteristics: under 75 years’ old, female, smokers, with right lung cancer (middle or lower lobe), and characterized by pathological stage I [62]. Lokowki [45] concluded that in NSCLC patients, elevated PLR (platelet–lymphocyte ratio) values appear to be an independent prognostic factor for survival. We can assert that in lung cancer patients, there is a complete re-organization of the immune system, particularly an increase in platelets and neutrophils; conversely, lymphocytes tend to decrease with an inverse dNLR or PLR.

The SII (systemic immune-inflammation index) or ALI (advanced lung inflammation index) are other parameters able to check the inflammatory status. A recent report [15] confirms the role of this indirect index in the prognosis of resected lung cancer. The SII and ALI have been related to the prognosis of other cancers; Fournel [63] showed the strong impact of systemic inflammation on the prognosis of malignant pleural mesothelioma and a shorter survival associated with the NLR, SII, and a lower ALI.

We can therefore assert that in patients affected by lung cancer, there is an important re-organization in the immune system, particularly an increase in platelets and neutrophils; contrarily, lymphocytes tend to decrease with a subversion of the different inflammatory parameters, such as the NLR, PLR, HALP, SII, and ALI. All these literature data demonstrate the strong association between the different inflammatory indexes analyzed, while the poor prognosis clearly corroborates the other literature data.

### 3.3. Interaction between General, Metabolic, Nutritional, and Inflammatory Status

BMI, weight loss, low muscle mass, and sarcopenia are closely linked with metabolism and the patient’s immune status and the explanation could reside in the metabolism of lung tumors and of the host; the absence of body reserves occurring in some lung cancer types inducing cachexia, could trigger a catabolic state favoring cancer development and, in addition, it is largely known that malnutrition is a cause of immunodeficiency promoting cancer progression. Thus, sarcopenia and weight loss could be responsible for immunity dysregulation, a catabolic state, and the consequent cancer promotion. On the other hand, more fat may encourage a good functionality of immune defenses against NSCLC. Some reports have shown how tumor-infiltrating CD8-lymphocytes are correlated with good nutritional status and associated with better survival [5], probably due to the promoting action given by the obesity/overweight condition in NSCLC to CD8+ lymphocytes, instead of an exhausting action.

From a metabolic point of view, tumor consumption could oblige the body to re-organize its metabolism; in particular, we refer to an enhancement of liver gluconeogenesis which assures more glucose. Gluconeogenesis is associated with lipolysis, which delays proteolysis. However, tumor development induces proteolysis too, particularly from amino acid consumption. The latter biochemical process is closely linked to the loss of skeletal muscle and assists in a vicious cycle of losing fat and muscle mass. The host’s metabolism is purposedly altered for producing glucose, facilitating cancer cell proliferation; they may prefer anaerobic glycolysis—Warburg effect—and this aspect might be associated with poor differentiation and survival [64,65,66]. Lactates resulting from anaerobic glycolysis (Warburg effect) could alter the immune response because they could influence the uptake of glucose by the tumor microenvironment cytotoxic cells. The effect could be a down-regulation of CD8+ cytotoxic T-cells in the local control of cancer progression.

Thus, the inflammatory status (pre-existing or concomitant to lung cancer), with subsequent increased energy consumption, contributes to malnutrition. In addition, catabolic processes secondary to inflammation, imbalance in anabolic and proteolytic pathways and reduced caloric intake, may be responsible for fat and muscle loss. In cancer patients, weight loss and sarcopenia are associated with physical decline, loss of quality of life, increased toxicity from radiation therapy and chemotherapy, and shorter survival. The combination of sarcopenia and cachexia is believed to be the direct cause of death in at least 20% of oncologic patients with advanced disease [9].

### 3.4. Prognostic Significance of Inflammation in Patients Undergoing Immune Checkpoint Inhibitor (ICI) Therapy

Immune checkpoint inhibitors (ICIs) have now become a cornerstone of clinical practice in oncologic treatment and they have transformed the therapeutic strategies of various cancers, including non-small cell lung cancer [67,68]. They exploit the inhibition of checkpoints, used by tumoral cells in order to defend themselves from immune system attacks.

Through interaction with the PD-1 receptor and PD-1 ligand 1 (PD-L1), ICIs take on a key regulatory role in T-cell activities; an upregulation of PD-L1 on the cell surface may inhibit T-cells from attacking, reduce FAS and interferon-dependent apoptosis, and protect cells from cytotoxic molecules produced by T-cells [69].

It is therefore obvious that inflammatory status plays a pivotal role in the real function and efficacy of immunotherapy. Indeed, in patients treated with ICIs, it is essential to evaluate the inflammatory status, before and during treatment [70].

As already debated in this paper, cancer genesis and metastasis are associated with systemic inflammation and malnutrition or a combination of these two aspects.

A recent metanalysis [70] brought to light how systemic inflammatory status may influence the effectiveness of ICIs and modify itself during therapy.

In particular, the authors detected several inflammatory indexes over-/under-expressed in patients, during medical (chemotherapy or immune therapy) treatment. Some of these indexes have been largely described in this paper; for example, albumin and neutrophil-to-lymphocyte ratio (NLR), have been shown to be able to predict the efficacy of ICI therapy in cancer patients [71,72]; the advanced lung cancer inflammation index (ALI), including albumin, NLR, and BMI, might help to predict survival outcomes in various tumors [73,74]. Another index under consideration is the ”Gustave Roussy Immune (GRIm) score”, which combines three blood markers: NLR, albumin, and lactate dehydrogenase (LDH). Depending on these factors, patients can be categorized into high-risk (score > 1) and low-risk groups (score > 1) [75,76,77]. The predictive and prognostic value of the GRIm score has been already studied in patients affected by advanced NSCLC after systemic treatment (chemotherapy, EGFR-TKIs, or second-line immunotherapy); this score was then modified by Li et al. [78] for hepatocarcinoma (HCC-GRIm score) and turned out to have higher predictive power in recognizing the patients who potentially benefitted from immunotherapy.

Once again, indices (ALI, and GRIm score) deriving from the immune and nutritional status of the host can significantly influence cancer prognosis. Last but not least, Jiang et al. [70] corroborated the prognostic value of the “ALI” and “GRIm” score for recognizing and evaluating responsiveness to immunotherapy for cancer patients, suggesting that they should be integrated into the routine assessment of these patients.

### 3.5. Prognostic Significance of Metabolic, Nutritional, and Inflammatory Status during Systemic Therapy for NSCLC

Patients undergoing postoperative chemotherapy often experience nausea, anorexia, anxiety, and transient weight loss, necessitating increased protein and energy support. Albumin levels are indicative of nutritional status, while adequate hemoglobin synthesis requires iron, vitamin B12, folate, protein, and vitamin C. Poor nutrition can lead to anemia due to insufficient hemoglobin synthesis.

On the other hand, as already widely stated, inflammation (marked by cytokines like IL-1, IL-6, IL-8, TNF-α, and IFN-γ and by complex indices such as the NLR, HALP, PLR, and SII) plays a crucial role in tumor development and in its prognosis (Table 4).

Guy et al. [79] focused on the effects of nutritional status improvement in patients, undergoing postoperative chemotherapy, with stage II-III NSCLC. In particular, they investigated the role of omega-3 fatty acid supplementation (ω-3 PUFA) on nutritional status and inflammatory response during chemotherapy. As suggested by Kaya et al. [80], patients with a ω-3 PUFA-enriched diet had significantly less albumin loss postoperatively, indicating improved nutritional recovery. The authors showed an improvement in hemoglobin and albumin levels after 12 weeks of ω-3 PUFA supplementation compared to placebo during postoperative chemotherapy.

These findings are in accordance with another study conducted by Liang et al. [81], where patients receiving ω-3 PUFAs and soybean oil, compared to those receiving only soybean oil post-colorectal surgery, showed lower IL-6 and TNF-α levels. Similarly, Weiss et al. [82] observed decreased IL-6 and TNF-α levels with fish oil intervention, consistent with current findings. These latter papers indicate that the improvement of nutritional status can improve general and inflammatory status, suggesting its role as an adjuvant anti-inflammatory agent. With this view, the authors concluded that ω-3 PUFA supplementation during postoperative chemotherapy improves nutritional status and reduces inflammatory responses in stage II-III NSCLC patients.

A study conducted by Cehreli et al. [83] focused on inflammatory and nutritional serum markers in the prediction of chemotherapy outcomes and survival in advanced-stage NSCLC. The authors investigated the relationship between malnutrition, as indicated by a low Subjective Global Assessement (SGA) and BMI, and cachexia. In particular, SGA [84], a standard for assessing nutritional status in cancer patients, is strictly correlated with physical function and symptoms such as loss of appetite, dyspnea, fatigue, and diarrhea. In their observations, despite a mean BMI of 20.8 kg/m^2^ not indicating malnutrition by WHO standards, 80% of patients were malnourished as per the SGA standard, highlighting BMI’s insufficiency due to factors like fluid retention.

In addition, high serum albumin levels, reflecting nutritional status and the Glasgow Prognostic Score (GPS), were associated with increased survival, aligning with the data of other studies [85,86]. Prealbumin, with a shorter plasma half-life, is a sensitive marker for protein-energy status and correlated with survival.

Combining these latter aspects (weight loss, BMI, SGA and serum albumin levels) suggested that the Nutritional Risk Index (NRI) could represent an index to predict good or bad prognosis in patients undergoing chemotherapy, even if future validation of specific tests for cancer patients is needed [83].

Concerning inflammatory status, a high expression of C-reactive protein (CRP), IL-6, TNF-α, and IL-1β [87,88], and zinc and vitamin D deficiency [89,90] (influencing oxidative stress and proinflammatory cytokines) associated with reduced albumin, weight loss, and poor performance, are crucial in predicting chemotherapy outcomes and morbidity.

A recent randomized controlled trial with 106 lung cancer patients, conducted by Zhang [91], focused on the effects of nutritional support based on the dietary anti-inflammatory index on cancer-related fatigue in lung cancer patients undergoing chemotherapy. The authors observed that the anti-inflammatory diet improved fatigue, Patient-Generated Subjective Global Assessment (SGA), albumin concentrations, CRP levels, and more generally the nutritional status and the quality of life of these patients.

Mitsuyoshi et al. [92] investigated the prognostic value of the CRP-BMI score, derived from C-reactive protein (CRP) levels and the body mass index (BMI), for patients with locally advanced non-small cell lung cancer (NSCLC) treated with chemoradiotherapy. The authors evaluated, among others, the effects of serum CRP, serum albumin levels, commonly used to assess nutritional status, BMI (<18.5 kg/m^2^), sarcopenia (indicating muscle wasting), and the skeletal muscle index (SMI). They concluded that the CRP-BMI score, easily acquired in clinical practice, was a useful prognostic tool. Higher CRP-BMI scores were associated with worse OS and reduced the likelihood of receiving salvage treatment upon recurrence.

### 3.6. Possible Clinical Impact on the Metabolic and Nutritional Management of Patients

It is well known that a correct treatment of lung cancer requires a multidisciplinary approach. The importance of neoadjuvant or adjuvant oncological treatment, the appropriate surgical timing, and the significance of radiotherapy have always been considered. However, in this integrated management, considering what has been previously discussed, the metabolic and nutritional status of the patients must necessarily be considered. Proper management of the inflammatory state, which is closely linked to metabolic and nutritional status, is crucial in evaluating potential therapeutic success, both in patients affected by early- and advanced-stage lung cancer.

First and foremost, at the moment of first diagnosis, it is crucial to assess the patient’s overall metabolic, nutritional, and inflammatory status. For instance, it would be beneficial to measure all the markers we previously discussed (albumin, prealbumin, vitamin D, zinc, inflammatory cytokines, CRP, inflammation markers, and derived indexes), to categorize the patient into risk groups.

Subsequently, alongside therapeutical management (radiotherapy, chemotherapy, immunotherapy, surgery, or combination of them), it should be mandatory to correct various parameters, especially nutritional and metabolic ones, through supportive therapies or specific dietary regimens, provided by specialized nutritionists.

This would allow the patient to initiate or continue specific therapies, bolstering their metabolic and inflammatory systems, thereby providing them with additional resources.

Monitoring these markers would also serve as a valuable prognostic instrument during treatments, using them as tools to assess the progress of therapies and to understand their effects.

An additional advantage is the extreme simplicity of marker monitoring. Alongside the conventional imaging investigations (CT scan, PET scan, and MRI), a simple blood sample could indeed be extremely useful for the patients.

Finally, these markers could aid in stratifying patients for personalized treatment, before, during, or after conventional treatments.

## 4. Conclusions

Generally speaking, nutritional and inflammatory status impact the medium- and long-term prognosis of lung cancer patients.

The data evaluated in the literature allow us to corroborate the concept that a correct stratification of patients could be helpful to identify those at risk, before or during surgical or medical treatment; this stratification could allow for an improved preoperative general, functional, nutritional, and inflammatory status of these patients, in order to minimize possible treatment complications and improve long-term prognosis. In this view, interaction between thoracic surgeons, medical and pulmonary physicians, physiotherapists, and nutritionists is mandatory to guarantee lung cancer patients the best of care.

## Figures and Tables

**Table 1 biomedicines-12-01554-t001:** Research selection and inclusion/exclusion criteria.

Items	Specification
Date of search	20 March 2024
Databases and sources	PubMed/Medline, Scopus, Google scholar
Search terms used	((lung cancer))) AND (((systemic inflammation) OR (inflammation) OR (inflammatory status))) AND (((general status) OR (nutritional status))) In addition, we manually screened the reference lists of the found studies in order to include other significant articles
Timeframe	2000–2024
Inclusion/exclusion criteria	Inclusion criteria: original articles, review articles, metanalysis Exclusion criteria: non-English language, case reports, case series
Selection process	AM independently conducted the research; all authors contributed to the paper’s final version
Eligible studies for review purposes	Seventy-nine (79)

**Table 2 biomedicines-12-01554-t002:** Nutritional and metabolic risk factors associated with worse prognosis of NSCLC at the moment of diagnosis.

Metabolic and Nutritional Factors
BMI < 18
Weight loss
Malnutrition
Sarcopenia
Skeletal muscle tissue
Obesity (obesity paradox)

**Table 3 biomedicines-12-01554-t003:** Inflammatory status. Main inflammatory markers in the blood. In the right column are their alterations, considered as risk factors in the prognosis of NSCLC.

Markers	Blood Levels
CRP (C-reactive protein)	increased levels
Blood count	leukocytosis, neutrophilia, thrombocytopenia, lymphocytopenia
HALP Hemoglobin (g/L) × Albumin (g/L) × Lymphocyte (/L)/Platelet (/L)	decreased hemoglobin * decreased albumin * decreased lymphocytes * decreased platelet
dNLR (derived neutrophil–lymphocyte ratio)	increased neutrophiles/decreased lymphocytes
LMR (lymphocyte–monocyte ratio)	increased monocytes/decreased lymphocytes
PLR (platelet–lymphocyte ratio)	decreased lymphocytes/decreased platelet
SII (systemic immune-inflammation index)	decreased serum platelets * increased neutrophiles/decreased lymphocytes
ALI (advanced lung inflammation index)	serum albumin * BMI/NLR; BMI = weight (kg)/height (m)^2^
GRIm score (Gustave Roussy Immune score)	increased NLR * decreased albumin * increased LDH High-risk (score > 1) and low-risk (score > 1)

*: multiplied.

**Table 4 biomedicines-12-01554-t004:** Metabolic, nutritional, and inflammatory prognostic factors during systemic therapy for NSCLC.

Prognostic Factors	Main Alterations
Albumin levels	decreased
Inflammatory cytokines (IL-1, IL-6, IL-8, TNF-α, IFN-γ)	increased
C-reactive protein	increased
Zinc	decreased
Vitamin D	decreased
Sarcopenia	muscle wasting and reduction in skeletal muscle index (SMI)
Body mass index	<18.5 kg/m^2^
Glasgow Prognostic Score (GPS)	increased C-reactive protein/decreased albumin levels
Nutritional Risk Index (NRI)	weight loss, decreased BMI and decreased albumin levels
CRP-BMI score	increased C-reactive protein/decreased BMI
Subjective Global Assessment (SGA)	weight loss, muscle wasting, fat loss, anorexia, cachexia

## Data Availability

Not applicable.

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
