# Peer review of "How General and Inflammatory Status Impacts on the Prognosis of Patients Affected by Lung Cancer: State of the Art"

_biomedicines, 2024, doi:10.3390/biomedicines12071554_

Round 1

Reviewer 1 Report

Comments and Suggestions for Authors

The manuscript discusses general and inflammatory status impacts on the prognosis of patients affected by lung cancer. In the manuscript, epidemiological data on lung cancer can be added. Prevalence data on pulmonary cancer associated with systemic inflammation or nutrition can be given (if available). In the introduction session, the objective of the review should be clearly stated. This manuscript is a type of systematic review and requires any PROSPERO registration. If so, a registration number can be given. A total number of literature is available and the number of eligible documents (for review purposes), should be clear. PRISMA flow diagram must be given. What is the main outcome of the review, and should it be discussed in detail before the conclusion section? Overall the manuscript is well presented.

Author Response

REVIEWER 1

The manuscript discusses general and inflammatory status impacts on the prognosis of patients affected by lung cancer. In the manuscript, epidemiological data on lung cancer can be added. Prevalence data on pulmonary cancer associated with systemic inflammation or nutrition can be given (if available). In the introduction session, the objective of the review should be clearly stated. This manuscript is a type of systematic review and requires any PROSPERO registration. If so, a registration number can be given. A total number of literature is available and the number of eligible documents (for review purposes), should be clear. PRISMA flow diagram must be given. What is the main outcome of the review, and should it be discussed in detail before the conclusion section? Overall the manuscript is well presented.

Concerning the first point (epidemiological data on lung cancer) we add some impressive numbers of American Cancer society:

Lung cancer is actually the second most commonly diagnosed cancer in both men and women; nevertheless, it represents the first cancer-related death in males and females, causing, in 2020, more deaths than breast, colorectal, and prostate cancers combined. In 2023, the impressive number of new cases of lung and bronchus cancers in US was 238,340; the estimated deaths in the same year for lung and bronchus was 127,070. in other words, this impressive figure almost represents a health emergency, being the leading cause of death in both men and women.

Concerning the second point (prevalence data on pulmonary cancer associated with systemic inflammation or nutrition); unfortunately, at the moment, to our best knowledge, we do not have precise prevalence data on the association between metabolic and nutritional status and cancer development and deaths. We only have studies where the correlation on prognosis and survival of these patients is very clear, as highlighted in the text in the different paragraphs.

Concerning the third point, this is a narrative review; the aim was to illustrate these emerging themes in a simple and concise way. For this reason, we did not use any PROSPERO registration.

In this optic, we effectively did not use the PRISMA checklist.

We add in the table the number of eligible documents for review purposes as rightly suggested by the reviewer.

Concerning the fourth point we add two new paragraphs.

3.5 Prognostic significance of metabolic, nutritional and inflammatory status during systemic therapy for NSCLC

Patients undergoing postoperative chemotherapy, often experience nausea, anorexia, anxiety, and transient weight loss, necessitating increased protein and energy support. Albumin levels are indicative of nutritional status, while adequate hemoglobin synthesis requires iron, vitamin B12, folate, protein, and vitamin C. Poor nutrition can lead to anemia due to insufficient hemoglobin synthesis.

On the other hand, as already widely stated previously, inflammation (marked by cy-tokines like IL-1, IL-6, IL-8, TNF-α, IFN-γ and by complex index as NLR, HALP, PLR, SII) plays a crucial role in tumor development and in its prognosis.

Guy et al (83) focused on the effects of nutritional status improvement in patients undergoing postoperative chemotherapy for patients with stage II-III NSCLC. Particularly they investigated the role of omega-3 fatty acid supplementation (ω-3 PUFA) on nutri-tional status and inflammatory response during chemotherapy. As suggested by Kaya et al (84), patients with ω-3 PUFA-enriched diet had significantly less albumin loss post-operatively, indicating improved nutritional recovery. The authors showed an improving of hemoglobin and albumin levels after 12 weeks of ω-3 PUFA supplementation com-pared to placebo, during postoperative chemotherapy.

These findings are in according with another study conducted by Liang et al. (85) where patients receiving ω-3 PUFA and soybean oil, compared to those receiving only soybean oil post-colorectal surgery, showed lower IL-6 and TNF-α levels. Similarly, Weiss et al. (86) observed decreased IL-6 and TNF-α levels with fish oil intervention, consistent with current findings. These latter papers indicates that the improvement of nutritional status can improve general and inflammatory status, suggesting its role as an adjuvant anti-inflammatory agent. In this optic, authors concluded that ω-3 PUFA supplementation during postoperative chemotherapy, improves nutritional status and reduces inflam-matory responses in stage II-III NSCLC patients.

A study conducted by Cehreli et al (87) focused on inflammatory and nutritional serum markers in the prediction of chemotherapy outcomes and survival in advanced stage NSCLC. The authors investigated the relationship between malnutrition, as indi-cated by low Subjective Global Assessement (SGA) and BMI, and cachexia. Particularly SGA (88), a standard for assessing nutritional status in cancer patients, is strictly correlated with physical function and symptoms such as loss of appetite, dyspnea, fatigue, and diarrhea. In their observations, despite a mean BMI of 20.8 kg/m² not indicating mal-nutrition by WHO standards, 80% of patients were malnourished per SGA, highlighting BMI’s insufficiency due to factors like fluid retention.

In addition, high serum albumin levels, reflecting nutritional status and the Glasgow Prognostic Score (GPS), were associated with increased survival, aligning with data of other studies (89,90). Prealbumin, with a shorter plasma half-life, is a sensitive marker for protein-energy status and correlated with survival.

Combining these latter aspects (weight loss, BMI, SGA, serum albumin levels) they suggested that the Nutritional Risk Index (NRI) could represent an index to predict good or bad prognosis in patients underwent chemotherapy, even if future validation of specific tests for cancer patients is needed (87).

Concerning inflammatory status, high expression of C-reactive protein (CRP), IL-6, TNF-α, IL-1β (91,92) and zinc and vitamin D deficiency (93,94) (influencing oxidative stress and proinflammatory cytokines) associated with reduced albumin, weight loss, poor performance, are crucial in predicting chemotherapy outcomes, morbidity.

A recent randomized controlled trial with 106 lung cancer patients, conducted by Zhang (95), focused on effects of nutritional support based on the dietary an-ti-inflammatory index on cancer-related fatigue in lung cancer patients undergoing chemotherapy. The authors observed that the anti-inflammatory diet improved fatigue, Patient-Generated Subjective Global Assessment (SGA), albumin concentrations, CRP levels and more generally the nutritional status and the quality of life of these patients.

Mitsuyoshi et al (96) investigated the prognostic value of the CRP-BMI score, derived from C-reactive protein (CRP) levels and body mass index (BMI), for patients with locally advanced non-small cell lung cancer (NSCLC) treated with chemoradiotherapy. Authors evaluated, among others, the effects of serum CRP, serum albumin levels, commonly used to assess nutritional status, BMI (< 18.5 kg/m²), sarcopenia (indicating muscle wasting) and skeletal muscle index (SMI). They concluded that the CRP-BMI score, easily acquired in clinical practice, was a useful prognostic tool. Higher CRP-BMI scores were associated with worse OS and reduced the likelihood of receiving salvage treatment upon recurrence.

3.6 Possible clinical impact on the metabolic and nutritional management of patients

It is well known that a correct treatment of lung cancer requires a multidisciplinary approach. The importance of neoadjuvant or adjuvant oncological treatment, the ap-propriate surgical timing, and the significance of radiotherapy have always been con-sidered. However, in this integrated management, considering what was previously discussed, the metabolic and nutritional status of the patients must necessarily be con-sidered. Proper management of the inflammatory state, which is closely linked to metabolic and nutritional status, is crucial in evaluating the potential therapeutic success, both in patients affected by early and advanced-stage lung cancer.

First and foremost, at the moment of first diagnosis, it’s crucial to assess the patient's overall metabolic, nutritional, and inflammatory status. For instance, it would be bene-ficial to measure all the markers we previously discussed (albumin, prealbumin, vitamin D, zinc, inflammatory cytokines, CRP, inflammation markers, derived indexes), to categorize the patient into risk groups.

Subsequently, alongside the therapeutical management (radiotherapy, chemo-therapy, immunotherapy, surgery or combination of them), it should be mandatory to correct various parameters, especially nutritional and metabolic ones, through supportive therapies or specific dietary regimens, provided by specialized nutritionists.

This would allow the patient to initiate or continue specific therapies, bolstering their metabolic and inflammatory systems, thereby providing them with additional resources.

Monitoring these markers would also serve as a valuable prognostic instrument during treatments, using them as tools to assess the progress of therapies and to under-stand their effects.

An additional advantage is the extreme simplicity of markers monitoring. Alongside the conventional imaging investigations (CT scan, PET scan, MRI), a simple blood sample could indeed be extremally useful for the patients.

Finally, these markers could aid in stratifying patients for personalized treatment, before, during or after the conventional treatments.

Reviewer 2 Report

Comments and Suggestions for Authors

The review is well-organized and summarizes the impact of general and inflammatory status on the prognosis of patients with lung cancers. However, several minor points need to be addressed before publication.

1. Spell out acronyms on first use, eg. BMI, TNM, NSCLC,

2. Check citations carefully. Line 106 discussed the impact of weight loss but cited ref 28, which concluded that “Loss of muscle mass and function may predate clinically overt cachexia, underlining the importance of evaluating sarcopenia, rather than weight loss alone.” 

3. Different parameters on general and nutritional status are actually correlated with each other, such as sarcopenia and weight loss. The author should present their general relationships and compare their impact on the prognosis based on literatures.

4. For general status, the impact of comorbidities (presence of other medical conditions), age, and smoking history, should be discussed as well.

Author Response

REVIEWER 2

The review is well-organized and summarizes the impact of general and inflammatory status on the prognosis of patients with lung cancers. However, several minor points need to be addressed before publication.

  1. Spell out acronyms on first use, eg. BMI, TNM, NSCLC

Done

  1. Check citations carefully. Line 106 discussed the impact of weight loss but cited ref 28, which concluded that “Loss of muscle mass and function may predate clinically overt cachexia, underlining the importance of evaluating sarcopenia, rather than weight loss alone.” 

Thanks for your observation. The concept of sarcopenia following weight loss was implied in the sentence. That’s the reason of the reference; we added this concept in the sentence.

  1. Different parameters on general and nutritional status are actually correlated with each other, such as sarcopenia and weight loss. The author should present their general relationships and compare their impact on the prognosis based on literatures.

I agree with the author. In the paper we examined how nutritional status, sarcopenia, obesity, weight loss, distribution of fat and muscle tissue can influence development of lung cancer.

Sarcopenia:

Sarcopenia represents another cornerstone for general and nutritional status of pa-tients. Low muscle mass negatively impacts on the survival of operated lung cancer patients (19, 32). Low muscle mass was mostly observed in underweight and normal weight patients (66% and 39% respectively), less frequently in overweight patients (21%), and rarely in obese patients (9.5%) (20). This scarcity of sarcopenic obesity (SO) in stage I–III patients agrees with the report of Prado et al. (33) which recorded 8% (20/250) of SO in a subpopulation of non-metastatic cancer patients, much lower than figures (around 25%) usually recorded in advanced and metastatic disease, where low muscle mass is included in a process of cachexia linked to poor survival (34,35). It is noteworthy that muscle loss occurring in follow-up, negatively impacts on the survival of early-stage resected NSCLC cancer patients independently (36). However, it is impossible to say if sarcopenia is likely to be a direct determinant of prognosis, but it is an extremely powerful marker of patient frailty. More generally, in cancer patients sarcopenia is associated (if not in part responsible for) physical decline, loss of quality of life, increased toxicity from chemotherapy, and shorter survival. The combination of sarcopenia and cachexia are believed to be the direct cause of death in at least 20% of oncologic patients with advanced disease (9).

Obesity and BMI

The body distribution of fat and muscle is essential; many reports showed that BMI impacts on survival independently from TNM (20).  A 2014-report by Lancet shows how obesity may increase the risk of 13 different cancer types not including lung cancer (21). Nowadays the impact of obesity on the long-term survival of cancer patients is really debated; in some cases, we assist to an oxymoron, called “the obesity paradox” (18-20, 22-24) which demonstrates how patients with pre-surgery BMI <25 kg/m2 or >25 kg/m2 had a significant difference in 5-year survival. The “obesity paradox” of NSCLC may therefore apply to lung cancer risk oc-currence (25), possibly due to the lower concentration of polycyclic aromatic hydrocarbons in lung cells due to carcinogenic molecules retention in fat tissue (26); on the other hand, also higher BMI would have a protective effect on the post-treatment outcome of lung cancer.

Nutritional status

Systemic inflammation and impairment of nutritional status are frequent features of lung cancer (3,4). These two factors are strongly correlated in patients undergoing lung cancer surgery and both are related to scarce infiltration of tumor by immune cells (5). Inflammatory status (pre-existent or concomitant with lung cancer), with subsequent increased energy consumption, contributes to malnutrition. In turn, catabolic processes secondary to inflammation, and reduced caloric intake, may be responsible for fat and muscle loss (4,5). Other mechanisms have also been evoked, including an imbalance in anabolic and proteolytic pathways (8,9)………

nutritional status and general status are key-features in the development of tumor characteristics and prognosis in patients affected by non-small-cell-ling cancer (NSCLC). The body distribution of fat and muscle is essential; many reports showed that BMI impacts on survival independently from TNM (20).

interaction of all these aspects

BMI, weight loss, low muscle mass and sarcopenia are closely linked with metabo-lism and the patient’s immune status; the explanation could reside in the metabolism of lung tumors and of the host; the absence of body reserves occurring in some lung cancer types inducing cachexia, could trigger a catabolic state favoring cancer development; in addition, it is largely known that malnutrition is a cause of immunodeficiency promoting cancer progression. Thus, sarcopenia and weight loss could be responsible for immunity dysregulation, catabolic state and the consequent cancer promotion. On the other hand, high fat may encourage a good functionality of immune defenses against NSCLC. Some reports showed how tumor infiltrating CD8-lymphocytes infiltrating were correlated with good nutritional status and associated with better survival, probably due to the promoting action given by obesity/overweight condition in NSCLC on CD8+ lympho-cytes, instead of an exhausting action.

From a metabolic point of view, tumor consumption could oblige the body to re-organize its metabolism; in particular we assist to an enhancement of liver glucone-ogenesis to assure more glucose. Gluconeogenesis is associated to lipolysis that delays proteolysis. However, tumor development induces proteolysis too, particularly from amino-acids consumption. Latter biochemical process is closely linked to the loss of skeletal muscle and we assist to a vicious cycle of losing fat and muscle mass. The host’s metabolism is purposedly altered for producing glucose, requiring cancer cells prolif-eration; they could prefer anaerobic glycolysis – Warburg effect – and this aspect could be associated with poor differentiation and survival (68-70). Lactates resulted from anaerobic glycolysis (Warburg effect) could alter the immune response because they could influence the uptake of glucose by the tumor microenvironment cytotoxic cells. The effect could be a down-regulation of CD8+ cytotoxic T cells in the local control of cancer progression.

Thus, the inflammatory status (pre-existing or concomitant to lung cancer), with subsequent increased energy consumption, contributes to malnutrition. In addition, catabolic processes secondary to inflammation, imbalance in anabolic and proteolytic pathways and reduced caloric intake, may be responsible for fat and muscle loss. In cancer patients, weight loss and sarcopenia are associated with physical decline, loss of quality of life, increased toxicity from radiation therapy and chemotherapy, and shorter survival. The combination of sarcopenia and cachexia are believed to be the direct cause of death in at least 20% of oncologic patients with advanced disease

In addition, we add two new paragraphs into the discussion.

3.5 Prognostic significance of metabolic, nutritional and inflammatory status during systemic therapy for NSCLC

Patients undergoing postoperative chemotherapy, often experience nausea, anorexia, anxiety, and transient weight loss, necessitating increased protein and energy support. Albumin levels are indicative of nutritional status, while adequate hemoglobin synthesis requires iron, vitamin B12, folate, protein, and vitamin C. Poor nutrition can lead to anemia due to insufficient hemoglobin synthesis.

On the other hand, as already widely stated previously, inflammation (marked by cy-tokines like IL-1, IL-6, IL-8, TNF-α, IFN-γ and by complex index as NLR, HALP, PLR, SII) plays a crucial role in tumor development and in its prognosis.

Guy et al (83) focused on the effects of nutritional status improvement in patients undergoing postoperative chemotherapy for patients with stage II-III NSCLC. Particularly they investigated the role of omega-3 fatty acid supplementation (ω-3 PUFA) on nutri-tional status and inflammatory response during chemotherapy. As suggested by Kaya et al (84), patients with ω-3 PUFA-enriched diet had significantly less albumin loss post-operatively, indicating improved nutritional recovery. The authors showed an improving of hemoglobin and albumin levels after 12 weeks of ω-3 PUFA supplementation com-pared to placebo, during postoperative chemotherapy.

These findings are in according with another study conducted by Liang et al. (85) where patients receiving ω-3 PUFA and soybean oil, compared to those receiving only soybean oil post-colorectal surgery, showed lower IL-6 and TNF-α levels. Similarly, Weiss et al. (86) observed decreased IL-6 and TNF-α levels with fish oil intervention, consistent with current findings. These latter papers indicates that the improvement of nutritional status can improve general and inflammatory status, suggesting its role as an adjuvant anti-inflammatory agent. In this optic, authors concluded that ω-3 PUFA supplementation during postoperative chemotherapy, improves nutritional status and reduces inflam-matory responses in stage II-III NSCLC patients.

A study conducted by Cehreli et al (87) focused on inflammatory and nutritional serum markers in the prediction of chemotherapy outcomes and survival in advanced stage NSCLC. The authors investigated the relationship between malnutrition, as indi-cated by low Subjective Global Assessement (SGA) and BMI, and cachexia. Particularly SGA (88), a standard for assessing nutritional status in cancer patients, is strictly correlated with physical function and symptoms such as loss of appetite, dyspnea, fatigue, and diarrhea. In their observations, despite a mean BMI of 20.8 kg/m² not indicating mal-nutrition by WHO standards, 80% of patients were malnourished per SGA, highlighting BMI’s insufficiency due to factors like fluid retention.

In addition, high serum albumin levels, reflecting nutritional status and the Glasgow Prognostic Score (GPS), were associated with increased survival, aligning with data of other studies (89,90). Prealbumin, with a shorter plasma half-life, is a sensitive marker for protein-energy status and correlated with survival.

Combining these latter aspects (weight loss, BMI, SGA, serum albumin levels) they suggested that the Nutritional Risk Index (NRI) could represent an index to predict good or bad prognosis in patients underwent chemotherapy, even if future validation of specific tests for cancer patients is needed (87).

Concerning inflammatory status, high expression of C-reactive protein (CRP), IL-6, TNF-α, IL-1β (91,92) and zinc and vitamin D deficiency (93,94) (influencing oxidative stress and proinflammatory cytokines) associated with reduced albumin, weight loss, poor performance, are crucial in predicting chemotherapy outcomes, morbidity.

A recent randomized controlled trial with 106 lung cancer patients, conducted by Zhang (95), focused on effects of nutritional support based on the dietary an-ti-inflammatory index on cancer-related fatigue in lung cancer patients undergoing chemotherapy. The authors observed that the anti-inflammatory diet improved fatigue, Patient-Generated Subjective Global Assessment (SGA), albumin concentrations, CRP levels and more generally the nutritional status and the quality of life of these patients.

Mitsuyoshi et al (96) investigated the prognostic value of the CRP-BMI score, derived from C-reactive protein (CRP) levels and body mass index (BMI), for patients with locally advanced non-small cell lung cancer (NSCLC) treated with chemoradiotherapy. Authors evaluated, among others, the effects of serum CRP, serum albumin levels, commonly used to assess nutritional status, BMI (< 18.5 kg/m²), sarcopenia (indicating muscle wasting) and skeletal muscle index (SMI). They concluded that the CRP-BMI score, easily acquired in clinical practice, was a useful prognostic tool. Higher CRP-BMI scores were associated with worse OS and reduced the likelihood of receiving salvage treatment upon recurrence.

3.6 Possible clinical impact on the metabolic and nutritional management of patients

It is well known that a correct treatment of lung cancer requires a multidisciplinary approach. The importance of neoadjuvant or adjuvant oncological treatment, the ap-propriate surgical timing, and the significance of radiotherapy have always been con-sidered. However, in this integrated management, considering what was previously discussed, the metabolic and nutritional status of the patients must necessarily be con-sidered. Proper management of the inflammatory state, which is closely linked to metabolic and nutritional status, is crucial in evaluating the potential therapeutic success, both in patients affected by early and advanced-stage lung cancer.

First and foremost, at the moment of first diagnosis, it’s crucial to assess the patient's overall metabolic, nutritional, and inflammatory status. For instance, it would be bene-ficial to measure all the markers we previously discussed (albumin, prealbumin, vitamin D, zinc, inflammatory cytokines, CRP, inflammation markers, derived indexes), to categorize the patient into risk groups.

Subsequently, alongside the therapeutical management (radiotherapy, chemo-therapy, immunotherapy, surgery or combination of them), it should be mandatory to correct various parameters, especially nutritional and metabolic ones, through supportive therapies or specific dietary regimens, provided by specialized nutritionists.

This would allow the patient to initiate or continue specific therapies, bolstering their metabolic and inflammatory systems, thereby providing them with additional resources.

Monitoring these markers would also serve as a valuable prognostic instrument during treatments, using them as tools to assess the progress of therapies and to under-stand their effects.

An additional advantage is the extreme simplicity of markers monitoring. Alongside the conventional imaging investigations (CT scan, PET scan, MRI), a simple blood sample could indeed be extremally useful for the patients.

Finally, these markers could aid in stratifying patients for personalized treatment, before, during or after the conventional treatments.

  1. For general status, the impact of comorbidities (presence of other medical conditions), age, and smoking history, should be discussed as well.

I agree with this observation. We reflected during the writing of the paper whether to discuss these topics. As widely demonstrated in the literature, comorbidities, smoking and age negatively influence the prognosis of lung cancer. However the aim of the review was to focus directly on the role of nutritional and inflammatory state on the prognosis, at the moment of diagnosis of the patients. For these reasons we have deliberately neglected the role of these other factors. Our fear was that we would not focus the attention on catabolic or metabolic processes linked to nutritional and inflammatory status, losing attention in this “mare magnum” of informations.

Reviewer 3 Report

Comments and Suggestions for Authors

I am afraid I did not find the manuscript appropriate to be published in the current form. Obviously, the landmark of a review manuscript is being ornamented by quite informative figures, tables and sometimes flowcharts. Here, it seems that the data are not categorized justly. The most important tables that I think are missed here are: a table for commonly checked factors in lung cancer; a list of markers, ratios and indexes checked in lung cancer; the effects of markers on survival prognosis and therapy prediction before and after chemo commencing. In discussion section, the authors may discuss about the effect of these markers in therapy process and how they can be used to improve OS and quality life of the patients.

Author Response

REVIEWER 3

I am afraid I did not find the manuscript appropriate to be published in the current form. Obviously, the landmark of a review manuscript is being ornamented by quite informative figures, tables and sometimes flowcharts. Here, it seems that the data are not categorized justly. The most important tables that I think are missed here are: a table for commonly checked factors in lung cancer; a list of markers, ratios and indexes checked in lung cancer; the effects of markers on survival prognosis and therapy prediction before and after chemo commencing. In discussion section, the authors may discuss about the effect of these markers in therapy process and how they can be used to improve OS and quality life of the patients.

I sincerely thank the reviewer for the remarks.

We add three tables in the manuscript, where you can find the most important metabolic process associated with worst prognosis. In the second table, concerning inflammatory status, we marked the main alteration of the blood markers, considered as risk and risk factors on the prognosis of NSCLC. In the third we explain the impact of inflammatory and metabolic status during systemic therapies.

We add two new paragraphs into the discussion:

3.5 Prognostic significance of metabolic, nutritional and inflammatory status during systemic therapy for NSCLC

Patients undergoing postoperative chemotherapy, often experience nausea, anorexia, anxiety, and transient weight loss, necessitating increased protein and energy support. Albumin levels are indicative of nutritional status, while adequate hemoglobin synthesis requires iron, vitamin B12, folate, protein, and vitamin C. Poor nutrition can lead to anemia due to insufficient hemoglobin synthesis.

On the other hand, as already widely stated previously, inflammation (marked by cy-tokines like IL-1, IL-6, IL-8, TNF-α, IFN-γ and by complex index as NLR, HALP, PLR, SII) plays a crucial role in tumor development and in its prognosis.

Guy et al (83) focused on the effects of nutritional status improvement in patients undergoing postoperative chemotherapy for patients with stage II-III NSCLC. Particularly they investigated the role of omega-3 fatty acid supplementation (ω-3 PUFA) on nutri-tional status and inflammatory response during chemotherapy. As suggested by Kaya et al (84), patients with ω-3 PUFA-enriched diet had significantly less albumin loss post-operatively, indicating improved nutritional recovery. The authors showed an improving of hemoglobin and albumin levels after 12 weeks of ω-3 PUFA supplementation com-pared to placebo, during postoperative chemotherapy.

These findings are in according with another study conducted by Liang et al. (85) where patients receiving ω-3 PUFA and soybean oil, compared to those receiving only soybean oil post-colorectal surgery, showed lower IL-6 and TNF-α levels. Similarly, Weiss et al. (86) observed decreased IL-6 and TNF-α levels with fish oil intervention, consistent with current findings. These latter papers indicates that the improvement of nutritional status can improve general and inflammatory status, suggesting its role as an adjuvant anti-inflammatory agent. In this optic, authors concluded that ω-3 PUFA supplementation during postoperative chemotherapy, improves nutritional status and reduces inflam-matory responses in stage II-III NSCLC patients.

A study conducted by Cehreli et al (87) focused on inflammatory and nutritional serum markers in the prediction of chemotherapy outcomes and survival in advanced stage NSCLC. The authors investigated the relationship between malnutrition, as indi-cated by low Subjective Global Assessement (SGA) and BMI, and cachexia. Particularly SGA (88), a standard for assessing nutritional status in cancer patients, is strictly correlated with physical function and symptoms such as loss of appetite, dyspnea, fatigue, and diarrhea. In their observations, despite a mean BMI of 20.8 kg/m² not indicating mal-nutrition by WHO standards, 80% of patients were malnourished per SGA, highlighting BMI’s insufficiency due to factors like fluid retention.

In addition, high serum albumin levels, reflecting nutritional status and the Glasgow Prognostic Score (GPS), were associated with increased survival, aligning with data of other studies (89,90). Prealbumin, with a shorter plasma half-life, is a sensitive marker for protein-energy status and correlated with survival.

Combining these latter aspects (weight loss, BMI, SGA, serum albumin levels) they suggested that the Nutritional Risk Index (NRI) could represent an index to predict good or bad prognosis in patients underwent chemotherapy, even if future validation of specific tests for cancer patients is needed (87).

Concerning inflammatory status, high expression of C-reactive protein (CRP), IL-6, TNF-α, IL-1β (91,92) and zinc and vitamin D deficiency (93,94) (influencing oxidative stress and proinflammatory cytokines) associated with reduced albumin, weight loss, poor performance, are crucial in predicting chemotherapy outcomes, morbidity.

A recent randomized controlled trial with 106 lung cancer patients, conducted by Zhang (95), focused on effects of nutritional support based on the dietary an-ti-inflammatory index on cancer-related fatigue in lung cancer patients undergoing chemotherapy. The authors observed that the anti-inflammatory diet improved fatigue, Patient-Generated Subjective Global Assessment (SGA), albumin concentrations, CRP levels and more generally the nutritional status and the quality of life of these patients.

Mitsuyoshi et al (96) investigated the prognostic value of the CRP-BMI score, derived from C-reactive protein (CRP) levels and body mass index (BMI), for patients with locally advanced non-small cell lung cancer (NSCLC) treated with chemoradiotherapy. Authors evaluated, among others, the effects of serum CRP, serum albumin levels, commonly used to assess nutritional status, BMI (< 18.5 kg/m²), sarcopenia (indicating muscle wasting) and skeletal muscle index (SMI). They concluded that the CRP-BMI score, easily acquired in clinical practice, was a useful prognostic tool. Higher CRP-BMI scores were associated with worse OS and reduced the likelihood of receiving salvage treatment upon recurrence.

3.6 Possible clinical impact on the metabolic and nutritional management of patients

It is well known that a correct treatment of lung cancer requires a multidisciplinary approach. The importance of neoadjuvant or adjuvant oncological treatment, the ap-propriate surgical timing, and the significance of radiotherapy have always been con-sidered. However, in this integrated management, considering what was previously discussed, the metabolic and nutritional status of the patients must necessarily be con-sidered. Proper management of the inflammatory state, which is closely linked to metabolic and nutritional status, is crucial in evaluating the potential therapeutic success, both in patients affected by early and advanced-stage lung cancer.

First and foremost, at the moment of first diagnosis, it’s crucial to assess the patient's overall metabolic, nutritional, and inflammatory status. For instance, it would be bene-ficial to measure all the markers we previously discussed (albumin, prealbumin, vitamin D, zinc, inflammatory cytokines, CRP, inflammation markers, derived indexes), to categorize the patient into risk groups.

Subsequently, alongside the therapeutical management (radiotherapy, chemo-therapy, immunotherapy, surgery or combination of them), it should be mandatory to correct various parameters, especially nutritional and metabolic ones, through supportive therapies or specific dietary regimens, provided by specialized nutritionists.

This would allow the patient to initiate or continue specific therapies, bolstering their metabolic and inflammatory systems, thereby providing them with additional resources.

Monitoring these markers would also serve as a valuable prognostic instrument during treatments, using them as tools to assess the progress of therapies and to under-stand their effects.

An additional advantage is the extreme simplicity of markers monitoring. Alongside the conventional imaging investigations (CT scan, PET scan, MRI), a simple blood sample could indeed be extremally useful for the patients.

Finally, these markers could aid in stratifying patients for personalized treatment, before, during or after the conventional treatments.

Reviewer 4 Report

Comments and Suggestions for Authors

The paper is not well written and lacks its aim. It isn't very clear and does not convey the purpose at all. The authors must work extensively to improve and make it suitable for publication.

Comments on the Quality of English Language

Need to improve for sentence-making and paragraph setting

Author Response

REVIEWER 4

The paper is not well written and lacks its aim. It isn't very clear and does not convey the purpose at all. The authors must work extensively to improve and make it suitable for publication.

I sincerely thanks the reviewer for his opinion.

We have extensively reviewed the paper, we added two new paragraphs in the discussion section. We have extensively discussed about the role of nutrition in order to correct inflammatory status during systemic therapy; we suggested how these informations can impact the current clinical practice. We added also three tables, in order to simplify the paper.

We add three tables in the manuscript, where you can find the most important metabolic process associated with worst prognosis. In the second table, concerning inflammatory status, we marked the main alteration of the blood markers, considered as risk and risk factors on the prognosis of NSCLC. In the third we explain the impact of inflammatory and metabolic status during systemic therapies.

We add two new paragraphs into the discussion:

3.5 Prognostic significance of metabolic, nutritional and inflammatory status during systemic therapy for NSCLC

Patients undergoing postoperative chemotherapy, often experience nausea, anorexia, anxiety, and transient weight loss, necessitating increased protein and energy support. Albumin levels are indicative of nutritional status, while adequate hemoglobin synthesis requires iron, vitamin B12, folate, protein, and vitamin C. Poor nutrition can lead to anemia due to insufficient hemoglobin synthesis.

On the other hand, as already widely stated previously, inflammation (marked by cy-tokines like IL-1, IL-6, IL-8, TNF-α, IFN-γ and by complex index as NLR, HALP, PLR, SII) plays a crucial role in tumor development and in its prognosis.

Guy et al (83) focused on the effects of nutritional status improvement in patients undergoing postoperative chemotherapy for patients with stage II-III NSCLC. Particularly they investigated the role of omega-3 fatty acid supplementation (ω-3 PUFA) on nutri-tional status and inflammatory response during chemotherapy. As suggested by Kaya et al (84), patients with ω-3 PUFA-enriched diet had significantly less albumin loss post-operatively, indicating improved nutritional recovery. The authors showed an improving of hemoglobin and albumin levels after 12 weeks of ω-3 PUFA supplementation com-pared to placebo, during postoperative chemotherapy.

These findings are in according with another study conducted by Liang et al. (85) where patients receiving ω-3 PUFA and soybean oil, compared to those receiving only soybean oil post-colorectal surgery, showed lower IL-6 and TNF-α levels. Similarly, Weiss et al. (86) observed decreased IL-6 and TNF-α levels with fish oil intervention, consistent with current findings. These latter papers indicates that the improvement of nutritional status can improve general and inflammatory status, suggesting its role as an adjuvant anti-inflammatory agent. In this optic, authors concluded that ω-3 PUFA supplementation during postoperative chemotherapy, improves nutritional status and reduces inflam-matory responses in stage II-III NSCLC patients.

A study conducted by Cehreli et al (87) focused on inflammatory and nutritional serum markers in the prediction of chemotherapy outcomes and survival in advanced stage NSCLC. The authors investigated the relationship between malnutrition, as indi-cated by low Subjective Global Assessement (SGA) and BMI, and cachexia. Particularly SGA (88), a standard for assessing nutritional status in cancer patients, is strictly correlated with physical function and symptoms such as loss of appetite, dyspnea, fatigue, and diarrhea. In their observations, despite a mean BMI of 20.8 kg/m² not indicating mal-nutrition by WHO standards, 80% of patients were malnourished per SGA, highlighting BMI’s insufficiency due to factors like fluid retention.

In addition, high serum albumin levels, reflecting nutritional status and the Glasgow Prognostic Score (GPS), were associated with increased survival, aligning with data of other studies (89,90). Prealbumin, with a shorter plasma half-life, is a sensitive marker for protein-energy status and correlated with survival.

Combining these latter aspects (weight loss, BMI, SGA, serum albumin levels) they suggested that the Nutritional Risk Index (NRI) could represent an index to predict good or bad prognosis in patients underwent chemotherapy, even if future validation of specific tests for cancer patients is needed (87).

Concerning inflammatory status, high expression of C-reactive protein (CRP), IL-6, TNF-α, IL-1β (91,92) and zinc and vitamin D deficiency (93,94) (influencing oxidative stress and proinflammatory cytokines) associated with reduced albumin, weight loss, poor performance, are crucial in predicting chemotherapy outcomes, morbidity.

A recent randomized controlled trial with 106 lung cancer patients, conducted by Zhang (95), focused on effects of nutritional support based on the dietary an-ti-inflammatory index on cancer-related fatigue in lung cancer patients undergoing chemotherapy. The authors observed that the anti-inflammatory diet improved fatigue, Patient-Generated Subjective Global Assessment (SGA), albumin concentrations, CRP levels and more generally the nutritional status and the quality of life of these patients.

Mitsuyoshi et al (96) investigated the prognostic value of the CRP-BMI score, derived from C-reactive protein (CRP) levels and body mass index (BMI), for patients with locally advanced non-small cell lung cancer (NSCLC) treated with chemoradiotherapy. Authors evaluated, among others, the effects of serum CRP, serum albumin levels, commonly used to assess nutritional status, BMI (< 18.5 kg/m²), sarcopenia (indicating muscle wasting) and skeletal muscle index (SMI). They concluded that the CRP-BMI score, easily acquired in clinical practice, was a useful prognostic tool. Higher CRP-BMI scores were associated with worse OS and reduced the likelihood of receiving salvage treatment upon recurrence.

3.6 Possible clinical impact on the metabolic and nutritional management of patients

It is well known that a correct treatment of lung cancer requires a multidisciplinary approach. The importance of neoadjuvant or adjuvant oncological treatment, the ap-propriate surgical timing, and the significance of radiotherapy have always been con-sidered. However, in this integrated management, considering what was previously discussed, the metabolic and nutritional status of the patients must necessarily be con-sidered. Proper management of the inflammatory state, which is closely linked to metabolic and nutritional status, is crucial in evaluating the potential therapeutic success, both in patients affected by early and advanced-stage lung cancer.

First and foremost, at the moment of first diagnosis, it’s crucial to assess the patient's overall metabolic, nutritional, and inflammatory status. For instance, it would be bene-ficial to measure all the markers we previously discussed (albumin, prealbumin, vitamin D, zinc, inflammatory cytokines, CRP, inflammation markers, derived indexes), to categorize the patient into risk groups.

Subsequently, alongside the therapeutical management (radiotherapy, chemo-therapy, immunotherapy, surgery or combination of them), it should be mandatory to correct various parameters, especially nutritional and metabolic ones, through supportive therapies or specific dietary regimens, provided by specialized nutritionists.

This would allow the patient to initiate or continue specific therapies, bolstering their metabolic and inflammatory systems, thereby providing them with additional resources.

Monitoring these markers would also serve as a valuable prognostic instrument during treatments, using them as tools to assess the progress of therapies and to under-stand their effects.

An additional advantage is the extreme simplicity of markers monitoring. Alongside the conventional imaging investigations (CT scan, PET scan, MRI), a simple blood sample could indeed be extremally useful for the patients.

Finally, these markers could aid in stratifying patients for personalized treatment, before, during or after the conventional treatments.

I hope that with these corrections and additions the reviewer will be satisfied.

The paper was reviewed by native-English speaker.

Round 2

Reviewer 3 Report

Comments and Suggestions for Authors

The manuscript looks improved now. Some minor rectifications in my opinion are as below:

1- The number of tables starts with “n”; n1. n 2 and etc. The letter “n” may be removed.

2- Table number 2 is not clear in my version of manuscript.

3- There must be homogeneity between the markers status. Either they must be stated by up and down arrows or the words “increased or decreased”. I prefer the second choice.

4- What does asterisk in table number 3 indicate?

Author Response

Thanks for your consideration

1- The number of tables starts with “n”; n1. n 2 and etc. The letter “n” may be removed.

Done

2- Table number 2 is not clear in my version of manuscript.

The table shows the main nutritional and metabolic risk factors associated with worst prognosis of NSCLC at the moment of diagnosis

3- There must be homogeneity between the markers status. Either they must be stated by up and down arrows or the words “increased or decreased”. I prefer the second choice.

Done

4- What does asterisk in table number 3 indicate?

*: multiplied. We add it into the legend of the table.

Reviewer 4 Report

Comments and Suggestions for Authors

The paper is poorly written and looks like a report of some hospital proceedings. There are no challenges, shortcomings or future directions talked about within relevant sections. Overall a weak work and needs a lot of work before it could be considered for publication.

Comments on the Quality of English Language

The sentences lack the flow of content. One page contains too many paragraphs in many places within the manuscript. 

Author Response

The paper is poorly written and looks like a report of some hospital proceedings. There are no challenges, shortcomings or future directions talked about within relevant sections. Overall a weak work and needs a lot of work before it could be considered for publication.

I totally respect your opinion and thank you for reviewing my paper.

Regarding the first observation, the purpose of a narrative review is to bring back, to its current state, all the knowledge inherent to a topic. "Ca va sans dire" that it must necessarily report the “hospital proceedings”.

Concerning the other remarks and the “poorness” or “weakness” of the paper, we extensively reviewed the literature, including more of 90 relevant papers focusing on the topic. The aim of this paper is not to give future directions or shortcomings. It could instead be useful in highlighting some characteristics (all the inflammatory, metabolic or nutritional risk factors) that are sometimes overlooked or even misunderstood in the management of NSCLC or in the correct stratification of the patients.

Comments on the Quality of English Language

The sentences lack the flow of content. One page contains too many paragraphs in many places within the manuscript.

Thanks.

In your first revision, you rated the English language with one star (extensive editing of English language required).

At the time of the first revision, the paper had been already revised by native-english speaker.

In this second revision nothing has been changed in the wording of the previous sentences.

We only added new paragraphs. But now the rating of the same manuscript is “three stars”, even if there were no changes in the sentences wording

However, the paper was revised by a native-english speaker.

Concerning “too many paragraphs in many places”, I don't understand the point of the observation. The chapter 3 (state of art), has different paragraphs because the topic is varied and addresses all different subjects.

Round 3

Reviewer 4 Report

Comments and Suggestions for Authors

The authors have not improved the paper and shall submit again for a fresh review.

Comments on the Quality of English Language

The flow of sentences and paragraph lengths must be rectified for improved transition.